# Caffeine Intake among Undergraduate Students: Sex Differences, Sources, Motivations, and Associations with Smoking Status and Self-Reported Sleep Quality

**DOI:** 10.3390/nu14081661

**Published:** 2022-04-16

**Authors:** Aina Riera-Sampol, Lluis Rodas, Sonia Martínez, Hannah J. Moir, Pedro Tauler

**Affiliations:** 1Research Group on Evidence, Lifestyles and Health, Department of Nursing and Physiotherapy, Research Institute of Health Sciences (IUNICS), University of the Balearic Islands, 07122 Palma, Spain; ana.riera@uib.es; 2Health Research Institute of the Balearic Islands (IdISBa), 07120 Palma, Spain; 3Research Group on Evidence, Lifestyles and Health, Department of Fundamental Biology and Health Sciences, Research Institute of Health Sciences (IUNICS), University of the Balearic Islands, 07122 Palma, Spain; lluisrodas@hotmail.com; 4School of Life Sciences, Pharmacy and Chemistry, Faculty of Science Engineering and Computing, Kingston University London, Penryhn Road, Kingston upon Thames KT1 2EE, UK; H.Moir@kingston.ac.uk

**Keywords:** undergraduate college students, caffeine intake, motivations, smoking, sleep quality

## Abstract

Due to its stimulatory effects, caffeine is one of the most frequently consumed mood and behavior altering drugs. University students report using caffeine-containing products to enhance mood and performance or for a desire of alertness. The current study investigated caffeine consumption in university undergraduate students, and associations with smoking status, alcohol and cannabis consumption, fruit and vegetable consumption, and sleep quality. Motivations for caffeine intake were also ascertained. A total of 886 undergraduates aged 18–25 years from the University of the Balearic Islands participated in a cross-sectional survey. Caffeine was consumed by 91.1% of participants. Caffeine consumers were more likely to be female, smokers, and alcohol and cannabis consumers. Coffee was found to be the main source of caffeine intake in both men and women (48.9% of total caffeine intake). Higher percentages of women consumed coffee (56.4 vs. 42.1%, *p* = 0.01) and tea (40.3 vs. 19.8%, *p* < 0.001), whereas a higher percentage of men consumed energy drinks (18.0 vs. 7.4%, *p* < 0.001). Main motivations for caffeine intake were those related to cognitive enhancement. Caffeine intake was associated with poorer subjective sleep quality (*p* < 0.001). In conclusion, undergraduate students that were female and smokers reported higher caffeine intakes. Coffee was found as the main caffeine contributor, with higher contributions of tea in women and energy drinks in men. Universities should consider the implementation of health campaigns and educational programs to educate students of the risks of high caffeine consumption together with associated behaviors such as smoking, alcohol consumption and poor sleep quality to physical health and academic performance.

## 1. Introduction

Caffeine is one of the most frequently consumed mood and behavior altering drugs [1]. The stimulatory effects of caffeine together with its widespread presence in foods such as coffee, tea, and chocolate are important reasons to explain the high prevalence (around 80%) of caffeine consumption around the world [1]. Regarding caffeine sources, in most European countries, except for the United Kingdom and Ireland, coffee has been commonly found to be the major source for adults [2].

Data from the 2007–2012 National Health and Nutrition Examination Survey (NHANES), reported that consumption of caffeine in U.S. adults was on average, 169 mg/day [3]. In Western Europe, including Spain, the average daily intake of caffeine is as similar to the U.S. [2]. In addition, similar or slightly lower figures for caffeine intake have been found in university [4,5] and in college students [6], or in participants of a similar age [2], with coffee as the main caffeine source. However, to our knowledge, no study has reported caffeine intake in university students from a South European and Mediterranean country such as Spain. Such countries have different coffee preparations (“cortado” or “café con leche”), and significant different dietary characteristics including the schedule and the energy distribution of the meals, and different working and teaching schedules.

Demographic factors such as sex, together with habits such as alcohol consumption and tobacco use have been associated with caffeine intake [3]. Regarding sex, studies performed in university students, or in participants of a similar age, have reported higher prevalence of caffeine consumption in women than in men [6], or a higher daily caffeine intake, but only when expressed as per kg body weight [4], because when expressed as the absolute value, no differences have been observed [4,6]. Therefore, it would be of interest to determine whether differences are related to sex differences in weight or for other reasons.

Previous studies have reported higher caffeine intake in smokers than in non-smokers in general population [7,8] and in college students from the USA [6]. Furthermore, a higher probability for smokers to exceed the recommended caffeine intake was also found in university students from New Zealand [4]. Notably, smokers may consume more caffeine because smoking upregulates the main metabolic pathway of caffeine [9,10], which could indicate that smokers may need to drink more caffeine to experience similar effects as non-smokers. However, characteristics of the diet, such as fruit and vegetable consumption, or the use of substances such as cannabis have not been commonly considered in relation to caffeine intake.

The main reasons reported for use of caffeine-containing products are usually related to the stimulant effects of caffeine, including to reduce fatigue, increase alertness or to enhance physical performance [2]. Adolescents state they use caffeine to provide more energy, for the taste of the product, and for image enhancement [11]. University students reported using caffeine-containing products to enhance mood and performance [6], or for a desire of alertness [12]. Furthermore, in a supplementation study, university students perceived that they were significantly more alert, awake, clear-minded, and able to concentrate after ingesting low doses of caffeine, with such effects being desirable for academic work [13].

Higher caffeine intake has been commonly associated with poorer subjective sleep quality in adults [14]. Furthermore, a higher prevalence of poor sleep quality has been found among consumers of energy and stimulant drinks [15]. However, most of the studies examined the association of subjective sleep quality with prevalence of caffeine sources’ consumption rather than with the overall daily caffeine consumption as a continuous variable. In addition, the influence of sex should be considered when determinants of subjective sleep quality are analyzed because a poorer subjective sleep quality in women than in men has been commonly found, at least among university students [16,17].

The aim of the study was to determine caffeine and caffeine sources’ consumption among undergraduate students from the University of the Balearic Islands. The motivations students reported for caffeine consumption were also ascertained. The association between caffeine intake and demographic variables such as sex and age, smoking status, and fruit and vegetable consumption were analyzed. Furthermore, the association of caffeine intake with subjective sleep quality was also analyzed. The main hypothesis of the study was that caffeine intake could be associated with smoking and with a poorer sleep quality.

## 2. Materials and Methods

### 2.1. Study Design and Participants

A descriptive cross-sectional study was performed in a convenience sample of university undergraduate students in March 2021. Participants could be included in the study if they were currently engaged in any undergraduate course in the University of the Balearic Islands and aged 18–26 years old. Participants completed, online and in a voluntary and completely anonymous fashion, a survey designed using the Google Forms web tool. The survey was distributed among university undergraduate students through announcements on the virtual learning environment, ‘Moodle’, which is an on-line educational platform used for academic purposes. Initially, 926 students completed the survey. However, 40 respondents were discarded due to incomplete questionnaires (related to independent variables: smoking (10), cannabis (20), and alcohol consumption (2)), or were outside the predefined age range (8). This lead to the final inclusion of 886 participants (about 7.4% of all potential participants, *n* = 11,926), where a minimum of 623 were required for significance (*p* < 0.05) based on the sampling population and to ensure a sample size large enough for the results being generalizable to all the undergraduate students from the University of the Balearic Islands [18].

All the participants were informed of the purpose and demands of the study. Consent to participate in the study was given by agreeing to complete the survey. The protocol was in accordance with the Declaration of Helsinki for research of human participants and was approved by the University of the Balearic Islands Research Ethics Committee (190CER21).

### 2.2. Data Collection

The following data were collected from participants using the online survey across eight sub-sections to determine caffeine intake (daily amount and sources), the motivations for caffeine intake and association with lifestyle factors such as fruit and vegetable intake, smoking status and alcohol consumption, as well as subjective sleep quality.

Sociodemographic variables. Information on sex, age and course year were collected.

Anthropometrical measurements. Self-reported mass and stature were recorded. Body mass index (BMI) was calculated as mass (kg) divided by stature (m) squared (kg·m^−2^).

Caffeine intake. Habitual daily caffeine intake was measured using a self-reported questionnaire previously developed and used by our group [19]. The frequency intake of common products containing caffeine was ascertained using the question “How often do you consume…?” with “daily” and “weekly” frequency options provided. Caffeine daily consumption was determined using the caffeine content of each product [2]. Products included in the questionnaire were coffee preparations, instant coffee, tea and mate, chocolate, cola drinks, and energy beverages. Furthermore, in an open question, participants were asked for any other caffeine source they commonly consumed. Results from a previous study performed in a university population demonstrated that the contribution to total caffeine intake of decaffeinated coffee was negligible (unpublished results). Therefore, this product was not considered in the caffeine intake questionnaire applied in the present study. Coffee intake was estimated from the intake of each coffee preparation contained in the questionnaire. The most common preparations in Spain were considered: espresso, “cortado” (espresso coffee, one serving, with a shot of milk), and “café con leche” (white coffee, or espresso coffee, one serving, with the remaining half of a cup filled with milk or steamed milk). Participants in the study were classified as either caffeine consumers or non-consumers. Participants reporting caffeine intake, any amount, from any source were considered as caffeine consumers.

Motivations for caffeine intake. A Spanish translation of the previously validated Caffeine Motives Questionnaire (CMQ) in participants with a similar age to those of the present study [20] was used to determine the motivations leading to caffeine consumption among participants. This questionnaire was comprised of 21 items, each one describing a reason for consuming caffeine. Questions were presented with a 5-point Likert scale (1: never ingest caffeine for this reason, 5: always ingest caffeine for this reason). Following the scoring instructions [20], a global score as well as four factors were calculated as the sum of all or some items. The first factor consisted of six items that reflected cognitive enhancement (alertness, concentration, drowsiness, attention, energy, to stay awake). The second factor was composed of three items that reflected negative affect relief (stress, anxiety, depression). The third factor was composed of nine items that reflected reinforcing effects (headache relief, taste, convenience, relaxation, social, craving, reward, cued craving, mood). The fourth factor was composed of three items that reflected weight control (ingredient in diet pills, powerful diuretic, to lose or control weight).

Alcohol consumption. Frequency of alcohol consumption was ascertained using the question (The Alcohol Use Disorders Identification Test (AUDIT)) “How often do you drink alcoholic beverages?” (Answers: never, once a month or less, 2–4 times a month, 2–3 times per week, and 4 or more times per week). Using this answer, participants were classified as either non-consumers (answered never), or alcohol consumers. For those participants reporting alcohol consumption, the following question was also included: “During a usual consumption day, how many servings of alcohol do you drink?”(Answers: 1–2, 3–4, 5–6, 7–9, 10 or more).

Smoking habit and cannabis consumption. Participants were asked if they smoked (YES/NO) in reference to tobacco consumption and cigarette use and were then classified as non-smokers, ex-smokers, occasional smokers or (daily) smokers, according to the criteria of the 2020 European Health Survey in Spain [21]. For most of the analyses performed in the present study, this variable was dichotomized into ‘smokers’ (daily and occasional smokers) and ‘non-smokers’ (ex-smokers and non-smokers). Furthermore, participants’ consumption of cannabis (YES/NO) was also ascertained.

Sleep quality. Participants’ self-reported sleep quality was measured using the short form of the MOS (Medical Outcomes Study) Sleep Scale [22], a six-item Likert-scale form that measures six sleep dimensions including: sleep initiation (time to fall asleep), sleep quantity, maintenance (e.g., middle and early morning awakening), respiratory problems or head ache, perceived adequacy, and somnolence (i.e., sleepiness or drowsiness) with the scale examples ranging from 1—“All of the time” to 5—“None of the time”. A correlation of 0.97 has been determined between the short and the long version [23]. The MOS Sleep Scale ranges from 6 to 30, and the higher the MOS Sleep Scale value is, the worse the subjective sleep quality is. The continuous score is considered as no cutoff points have been established [22].

Fruit and vegetable daily intake. The number of daily servings of fruit and vegetables consumed were determined as a marker of diet quality by self-reported average daily consumption with the question “How many servings of fruit/vegetable do you consume daily?” Examples of servings were provided in the survey as follows: 1 average dish of salad, 1 average dish of cooked vegetables, 2 cucumbers, 2 carrots, 1 medium-sized fruit (apple, orange, pear, banana), 1 slice of melon or watermelon, 2–3 small fruits (apricots, plums), or one cup of very small fruit (berries, grape).

### 2.3. Statistical Analysis

Statistical analysis was carried out using IBM SPSS Statistics 24.0 software (SPSS/IBM, Chicago, IL, USA). Statistical significance was accepted at a *p*-value below 0.05. To maintain an homogenous number of participants throughout the study and in all the analyses, only participants who provided all the information required were considered in the statistical analysis (*n* = 886). All the data were tested for normal distribution (Kolmogorov–Smirnov test). A Student’s *t*-test for unpaired data or Pearson’s chi-square (χ2) test was used to evaluate differences between sexes. Cohen’s d was determined as a measure of effect size for *t*-test comparisons. Descriptive analysis was used to report the frequencies and percentages of categorical variables, and the mean and standard deviation (SD) were reported for quantitative variables. The one-way ANOVA with Bonferroni correction was used to compare the intake of caffeine and coffee between participants stratified per smoking habit (daily, occasional, ex- and non-smokers). Eta squared (η^2^) was determined as a measure of effect size for the ANOVA. Multiple linear regression analysis, using the stepwise procedure, was applied to determine the association between the amount of daily caffeine intake (dependent variable) and independent and control variables (smoking (YES/NO), age, sex, cannabis consumption (YES/NO), alcohol consumption (YES/NO) and daily intake of servings of fruit and vegetables). Multiple regression analysis was also performed in participants distributed per sex. Logistic regression analysis was also used to determine the association between caffeine intake (YES/NO) and all independent variables. The same logistic regression model was used to determine the association between consumption of each caffeine source (YES/NO) and independent variables. Finally, multiple linear regression analysis was applied to determine the association between subjective sleep quality (dependent variable) and caffeine daily intake, with sex, age, smoking (YES/NO), cannabis consumption (YES/NO), alcohol consumption (YES/NO) and daily intake of servings of fruit and vegetables as control variables. Regression models for subjective sleep quality for men and women were also obtained.

## 3. Results

### 3.1. Characteristics of Participants in the Study

Of the participants in the study, 69% were women. This percentage is slightly higher than the proportion of female students in the university (about 59%). Table 1 shows the general characteristics of participants in the study as a whole and stratified by sex. For men, a significantly higher prevalence of cannabis consumption (*p* < 0.001) was found. However, the prevalence of alcohol consumption was higher among women (*p* = 0.034). Regarding diet, a higher consumption of fruit and vegetable servings were found in women (*p* = 0.010). Self-reported sleep quality was poorer in women than in men as indicated by the higher score in the MOS-sleep scale (*p* = 0.014).

### 3.2. Caffeine and Caffeine Sources’ Consumption among Participants in the Study

Average daily caffeine consumption for all students, including non-consumers, was 155.4 mg, with a mean intake of 172.5 mg among caffeine users (Table 2), which corresponds to the caffeine content in two cups of coffee. Women reported a higher caffeine intake than men, both expressed as the absolute intake (*p* = 0.007) and the intake per kg of body weight (*p* < 0.001). A higher daily intake than recommended (400 mg) [2], the content of 4–5 cups of coffee, was reported by 77 participants (9.5% of consumers), with similar percentages for men and women (9.0 vs. 9.8%). Caffeine, in any form, was consumed by 91.1% of participants, with higher figures in women than in men (*p* = 0.010). Regarding the sources of caffeine, coffee (*p* = 0.010), instant coffee (*p* = 0.019), and mainly tea and mate (*p* < 0.001) were consumed by higher percentages of women than men. However, a higher percentage of men consumed energy drinks (*p* < 0.001). Of interest, 128 participants (15.9% of caffeine consumers) indicated chocolate as their only source of caffeine.

When the contribution of each caffeine source to total caffeine intake was analyzed (Table 3), coffee was found to be the main source both in men and in women (about half of the total caffeine intake). The contribution of tea and mate was about twice as high in women than in men (*p* < 0.001). On the other hand, the contribution of energy drinks was about five times higher in men than in women (*p* < 0.001).

### 3.3. Analysis of the Variabales Associated with Caffeine and Caffeine Sources’ Consumption

Linear regression was used to examine daily caffeine intake (mg·day^−1^). Logistic regression was applied to examine whether participants consumed caffeine and caffeine sources or not. Table 4 shows the results of the multivariable regression analysis for daily caffeine intake (dependent variable). Smoking was found as the main predictor for a higher daily intake of caffeine (change in R^2^ 0.052, *p* < 0.001). A higher fruit and vegetable intake (*p* < 0.001), alcohol consumption (*p* < 0.001), cannabis consumption (*p* = 0.001), female sex (*p* = 0.014) and older age (*p* = 0.039) were also significantly associated with a higher caffeine intake.

Multivariate linear regression models for daily caffeine intake were also obtained for men (Appendix A) and women (Appendix A). Smoking remained as the main predictor in men and women, with similar standardized coefficients (0.227 vs. 0.153, respectively). In women, age became a non-significant predictor (*p* = 0.604), while similar coefficient values for fruit and vegetable intake and smoking were observed (0.166 vs. 0.153).

Table 5 shows results of the logistic regression analysis for caffeine intake (YES/NO). This analysis revealed that caffeine consumers were more likely to be women (*p* = 0.022), smokers (*p* = 0.048) and alcohol consumers (*p* = 0.002).

The logistic regression analysis was also applied to the consumption of caffeine sources considered: coffee, instant coffee, tea and mate, chocolate, cola drinks, and energy drinks. All variables included in the analysis, except cannabis consumption, were significantly associated with consuming coffee (Appendix A). However, consuming instant coffee was only associated to female sex. Tea and mate intake was associated with female sex and with a higher intake of fruit and vegetables (Appendix A). Cola drinks’ intake was found to be associated with smoking and with alcohol consumption (Appendix A). Energy drinks’ intake was associated with male sex and being a smoker.

### 3.4. Caffeine Consumption Stratified per Smoking Habits

Significant differences (*p* < 0.001) were found for daily caffeine intake between participants in the study distributed by the smoking categories (Table 6). No differences were observed between non-smokers and ex-smokers (*p* = 1.00), neither between occasional and habitual smokers (*p* = 1.00). A higher caffeine intake was observed in habitual smokers than in ex- (*p* = 0.030) and non-smokers (*p* < 0.001). A similar picture was observed for coffee intake (expressed as the number of coffee servings consumed daily), with significant differences between categories (*p* < 0.001) and higher values in smokers than in non-smokers.

Table 7 shows the pattern of caffeine and caffeine sources in non-smokers and smokers. There were significantly higher values for daily caffeine (*p* < 0.001) and coffee (*p* < 0.001) intake in smokers than in non-smokers. Furthermore, the prevalence of consumption of caffeine (*p* = 0.006), coffee (*p* < 0.001), cola drinks (*p* < 0.001), and energy drinks (*p* < 0.001) were higher among smokers.

### 3.5. Multivariate Regression Analysis for Subjective Sleep Quality

Table 8 shows results of the regression analysis for sleep quality as a dependent variable, with higher values of the scale being associated with a worse subjective sleep quality. Caffeine intake was found as the main predictor, with a negative effect on sleep quality (*p* < 0.001). Furthermore, a higher daily intake of fruit and vegetables (*p* = 0.001) and male sex (*p* = 0.030) were associated with better sleep quality. No associations with sleep quality were found for age, smoking and cannabis consumption.

Multivariate linear regression models for subjective sleep quality were also obtained for men (Appendix A) and women (Appendix A). Caffeine and fruit/vegetable intake remained as the only significant predictors for subjective sleep quality in men and women.

### 3.6. Motivations for Caffeine Consumption among Participants in the Study

No differences between men and women were found for the global CMQ score (Table 9). Regarding the four factors considered, higher values for women than for men were found for Factor 4 (*p* = 0.016), which included motives related to weight control. Considering the factors’ score ranges (lowest and highest possible score), and the participants’ scores, the most relevant factor was Factor 1 (cognitive enhancement). While this Factor 1 reached about 39% of its highest score, the other factors reached between 3% and 15% of the score range (Appendix A).

When the items included in the questionnaire were considered (Appendix A), those with the highest scores were related to the “like of the taste of caffeinated beverages” (3.32 ± 1.41) and to “stay awake” (3.23 ± 1.46). Regarding differences between men and women, significantly higher values in women were found for the reasons “to feel more alert” (*p* = 0.011), “as a reward to myself for completing a task” (*p* = 0.008), and “because it is a powerful diuretic” (*p* = 0.022). On the other hand, a higher value in men was observed for the reason “because it’s convenient to drink caffeinated beverages” (*p* = 0.028).

## 4. Discussion

The aim of the present study was to determine the pattern of caffeine and caffeine sources’ consumption and the motivations for caffeine consumption among undergraduate students from the University of the Balearic Islands. The main results of the present study are the differences found between men and women, and also between smokers and non-smokers, regarding the pattern of caffeine consumption. Undergraduate university female students reported a higher daily consumption of caffeine, with higher prevalence of coffee and tea intake. On the other hand, men reported a higher prevalence of energy drink consumption. Higher caffeine consumption was associated with habits such as smoking and alcohol consumption, as well as with a higher intake of fruit and vegetables. Caffeine intake was also associated with poorer subjective sleep quality.

Caffeine was consumed by 91.1% of participants in the present study. This value was very similar to studies performed in university or college students from the US [6] or Netherlands [5]. However, it should be noted that in the present study, about 16% of consumers reported caffeine intake only from chocolate, a caffeine source that was not always considered in previous studies. In spite of this observation, the average daily caffeine consumption observed was similar to that found in previous studies focused on university students from different locations such as the US [6], New Zealand [24], Netherlands [5] and the European surveys examining similar age groups [2]. Furthermore, measured daily caffeine intake was also similar to a previous study performed by our research group in a cohort of slightly older participants from the same university (155.4 mg in the present vs. 164.3 mg in the previous study) [19]. Among caffeine consumers, 9.5% of participants in the present study reported daily caffeine intakes higher than the recommended amount (400 mg) [2]. This percentage is slightly lower than that recently found in a similar population from New Zealand (15%) [4], where the prevalence of caffeine consumption was about 99%, but this was higher than values usually found in even older populations [2,25]. Due to the potential negative effects of excessive caffeine intake, future studies should pay attention on these high consumers to prevent potential public health issues [4].

In the present study, prevalence of caffeine consumption observed among women was higher than in men [6], with also a higher daily caffeine intake per kg of body weight [4]. It has been reported that caffeine in women is metabolized 20–30% faster than in males [26]. Therefore, women could be able to consume relatively higher amounts of caffeine without experiencing more intense or prolonged effects [4].

A weak association was found between caffeine consumption and age. This association has been previously reported in college students [6], and becomes stronger when a wider age range is considered [3]. Reasons for this association are not well-understood, and it has been mainly attributed to increasing demands over the lifespan rather than genetic or physiological reasons [3].

In agreement with previous studies performed in similar populations, the source that most contributed to total caffeine intake was coffee [5,6,27], with a higher prevalence of coffee consumption in women than in men [6]. The current study also noted a lower consumption of tea compared with studies performed in other countries [4,6], which may be due to cultural differences in drink preferences. The prevalence of tea (and mate) intake, as well as the contribution of tea to total caffeine intake, was, however, higher in women than in men, which agrees with previous observations in college students [6]. On the other hand, as has been previously observed [6,28], higher values of consumption and contribution to total caffeine intake for energy drinks were found in men. In spite of the increasing popularity [29], the highest consumption commonly observed among young people [25,27,30], including university students [6,28], energy drinks accounted for less than 4% of the total caffeine consumption, resulting in the lowest contributor among caffeine sources considered. Furthermore, prevalence of consumption (10.7%) was lower than in previous studies, with values as high as 23% [29] or one-third of college students consuming at least one energy drink over the previous month [29]. Finally, regarding caffeine sources, the high contribution of chocolate, as the second main source, should be highlighted. This is of special interest because similar previous studies focused on university students considered only caffeinated beverages as caffeine sources [5,6,12]. Therefore, results from the present study provides novel insight into the main caffeine sources in university students.

Previous studies have reported higher caffeine intakes among smokers than non-smokers [6,31]. In agreement with these observations, results from both the multivariate and the logistic regression analyses showed an association between caffeine intake and smoking. Furthermore, results from the present study suggest that this association depends on the caffeine source, as the prevalence of coffee, cola and energy drink consumption was found to be higher among smokers, while no differences were observed for the remaining caffeine sources considered. These results are similar to previous studies showing that smokers consumed more coffee [6,12], soda [6] and energy drinks [6] than non-smokers. Physiological, cognitive and environmental factors may all contribute to the association between smoking and caffeine intake. Smoking increases the rate of caffeine metabolism [9,10]; as a consequence, smokers must consume caffeine more frequently than non-smokers to maintain similar effects. Other factors, such as stress, could have similar effects on the use of both caffeine and nicotine leading to increased use of both [32]. A previous study performed in a young population showed that nearly 50% of participants reported increasing caffeine consumption when under stress [33].

Higher caffeine intake has been also associated with alcohol consumption, a result commonly described in the literature [3]. This association has been attributed, at least in part, to the sharply increased combined use of caffeine and alcohol, with studies suggest that such combined use may increase the rate of alcohol-related injury [34]. In agreement with this suggestion, in the present study a significant association was found for alcohol and cola drinks consumption. Caffeine consumption was also associated to cannabis consumption. The main reason for this association remains unknown, and cannabis use has been more linked to alcohol and tobacco consumption [35]. Biological effects, such as cross-sensitization and activation of the reward system, or because the use of one substance influences the metabolism or enhances the effect of another substance cannot be discarded [35].

On the other hand, a positive association was found in the current study between caffeine intake and servings of fruit and vegetables. This result is contrary to those found in previous studies reporting associations between caffeine, or coffee intake, and negative characteristics of the diet, including low vegetable intake [36]. However, it should be considered that these studies were performed in older populations, with coffee consumers reporting higher prevalence of hypercholesterolemia and other conditions that could involve dietary limitations, while similar studies focused on caffeine consumption in university or college students or in younger populations did not consider characteristics of the diet [4,5,6,12].

Examining factors associated with subjective sleep quality in university students is essential, as it has been suggested that risk factors for poor sleep quality in young adulthood are likely to differ from risk factors in adolescence or adulthood [37]. Increased consumption of caffeinated beverages has been associated with poorer subjective sleep quality in university students [12,38,39]. In agreement with these observations, in the present study, caffeine intake was found as the main predictor of poorer subjective sleep quality. Proper sleep homeostasis has been related to effects of high adenosine concentrations during time awake, and their dissipation during the sleep episode [40]. Caffeine, due to its adenosine receptor antagonistic nature [41], could interfere with this mechanism, leading to sleep disruptions and worse sleep characteristics. Furthermore, it has been reported that caffeine may inhibit adenosine from melatonin secretion [42]. Melatonin has been considered as a key neurohormone in the regulation of the sleep-awake cycle in humans, with levels starting to rise about two hours prior to habitual bedtime [43]. Therefore, caffeine intake could modify the common circadian rhythm of melatonin, leading to sleep disturbances. Previous studies also reported worse values for different parameters related to sleep quality in female than in male students [16,17]. This is similar to the findings of the present study, showing worse subjective sleep quality in women. Poorer sleep quality in female than in male university students has been related to the lower stress levels in male students [44,45]. For instance, personal and academic stress have been reported to negatively affect sleep quality in college students [44]. However, it should be considered that, in addition to the aforementioned parameters, there may be other factors that significantly affect sleep quality such as stress, electronic device usage, physical activity, and fast-food consumption [12]

In the present study, subjective sleep quality was found to be inversely associated with fruit and vegetable consumption. In university and college students, an inverse association between fruit and vegetable consumption with poor sleep quality and restless sleep was reported [46]. It has been suggested that the positive effects of fruit and vegetable intake on sleep may be due to the high content of melatonin and serotonin [47], but also to the high amount of antioxidants, which could play a role in improving sleep quality [48,49]. On the other hand, poorer sleep quality is associated with higher intake of energy-dense foods but lower intake of fruit and vegetables [49]. Furthermore, sleep disruption could modify the pattern of appetite-related hormones, leading to an increase of energy-dense foods and a decrease in fruit and vegetable consumption [50]. Therefore, the relationship between fruit and vegetable intake and sleep parameters has been suggested to be bidirectional [51]. Taking into account the nature of this relationship, public health interventions, or approaches from university health services, should be addressed to increase fruit and vegetable intake, but also to improve sleep hygiene [49].

Factors considered in the present study (sex, age, smoking, alcohol and cannabis consumption, and diet quality) could explain only a limited fraction of all inter-individual variation in caffeine consumption. To complete this picture, participants’ reasons to consume caffeine were also ascertained. Such reasons related to staying awake, to avoid drowsiness, to feel more alert, to enjoy taste, and to help in concentration, were among those with the highest scores, being, some of them of interest for university students to enhance their academic performance. These results were very similar to those reported in college students from five universities in the US [6], in which most participants reported using caffeine-containing products to enhance some aspect of mood or performance, mainly to feel more awake and alert, and to enjoy the taste, as well as to improve concentration. These observations agree with results from research studies reporting positive effects of caffeine intake on mood and cognitive performance [52,53,54]. However, it was also shown that if the caffeine dose was too high (500 mg), the arousal levels of the participants were overstimulated, and continuous attention task performance began to decline [55]. It is noteworthy that, within the low scores observed in the present study, higher values in women were found for the factor related to weight control (Factor 4). Analyzing scores of each item included in this factor, the difference can be mainly attributed to the use of caffeine as a diuretic, but also to help lose or control body weight. A previous study showed that caffeine consumption was significantly associated with weight loss [6], but differences between sexes were not analyzed.

This study presents some limitations that should be acknowledged. In addition to the limitations due to the observational nature, the current study was limited to cross-sectional self-reported data from students belonging to a single Balearic Spanish university. Furthermore, the different questionnaires used in the present and in previous studies to record caffeine intake could have influenced comparisons with previous studies. Motivations for caffeine consumption were ascertained in general rather than for each caffeine containing product, which could have provided valuable information, because a previous study performed in university students suggested that the main reasons for consuming each product could be different [12]. Regarding sleep quality, factors that could affect sleep such as noise, stress, artificial light or electronic media usage were not considered in the present study. Furthermore, specific sociodemographic variables such as socioeconomic status were not assessed. The survey was performed when some restrictions due to COVID-19, for example limited opening times of cafes, bars or restaurants, were still in effect, which could have affected the answers of participants. Finally, data collection took place at the beginning of the second term within the academic calendar, which supposes, in general, a non-stressful academic period. It could be expected that patterns determined in the present study change in more demanding academic periods.

## 5. Conclusions

Women and smokers of university undergraduate students reported higher caffeine intakes. The faster metabolism of caffeine in both women and smokers could be the main reason leading to this higher intake. Coffee was found as the main caffeine contributor, with higher contributions of tea in women and energy drinks in men. Caffeine intake was also found to be associated to poor subjective sleep quality. Main reasons for caffeine intake among university students were related to increased mood and alertness, together with enjoying the taste. Universities should consider the implementation of health campaigns and educational programs to educate students of the risks of high caffeine consumption and poor sleep quality to physical health and academic performance. Associations reported in the present study could allow implementing appropriate educational strategies to address behaviors in combination such as smoking, alcohol consumption and excessive caffeine intake, all of which are linked to an increased risk of poor health. Furthermore, results obtained could help universities and health professionals working with students to increase sleep quality and, potentially, academic performance.

## Figures and Tables

**Table 1 nutrients-14-01661-t001:** General characteristics of participants in the study.

	All (*n* = 886)	Men (*n* = 278)	Women (*n* = 608)	*p* Value (Cohen’s d)
**Age (years)**	20.6 ± 2.1	20.6 ± 2.1	20.6 ± 2.1	0.705 (−0.027)
**Course year**				0.036 *
First	283 (31.9)	105 (37.8)	178 (29.3)	
Second	261 (29.5)	80 (28.8)	181 (29.8)	
Third	195 (22.0)	58 (20.9)	137 (22.5)	
Fourth	147 (16.6)	35 (12.6)	112 (18.4)	
**Body mass (kg)**	64.2 ± 12.8	72.7 ± 11.7	60.3 ± 11.4	<0.001 * (1.074)
**Stature (cm)**	169 ± 9	178 ± 8	165 ± 7	<0.001 * (1.944)
**BMI (kg·m^−2^)**	22.5 ± 3.7	22.9 ± 3.2	22.3 ± 3.9	0.008 * (0.180)
**Smoking**				0.180
Daily smokers (*n* (%))	73 (8.2)	29 (10.4)	44 (7.2)	
Occasional smokers (*n* (%))	52 (5.9)	13 (4.7)	39 (6.4)	
Ex-smokers (*n* (%))	55 (6.2)	13 (4.7)	42 (6.9)	
Non- smokers (*n* (%))	706 (79.7)	223 (80.2)	483 (79.4)	
**Cannabis consumption**				
Yes (*n* (%))	39 (4.4)	19 (6.8)	20 (3.3)	0.017 *
**Alcohol consumption**				
Yes (*n* (%))	680 (76.7)	201 (72.3)	479 (78.8)	0.034 *
**Fruit and vegetables intake**				
Fruit (servings·day^−1^)	1.72 ± 1.26	1.67 ± 1.32	1.74 ± 1.24	0.465 (−0.053)
Vegetables (servings·day^−1^)	1.52 ± 1.19	1.30 ± 1.04	1.62 ± 1.25	<0.001 * (−0.271)
Fruit and vegetables (servings·day^−1^)	3.24 ± 2.08	2.97 ± 1.96	3.36 ± 2.12	0.010 * (−0.187)
**Sleep quality**	14.3 ± 3.8	13.9 ± 3.6	14.5 ± 3.9	0.014 * (−0.173)

Values are the mean ± SD or number of participants (percentage). * *p* < 0.05 indicates significant differences between men and women, as determined by Student’s *t*-test for unpaired data or Pearson’s chi-square (χ2). Cohen’s d is provided as a measure of effect size for *t*-test comparisons.

**Table 2 nutrients-14-01661-t002:** Caffeine intake and prevalence of consumption of caffeine and caffeinated products.

	All (*n* = 886)	Men (*n* = 278)	Women (*n* = 608)	*p* Value (Cohens’ d)
Caffeine (mg·day^−1^)	155.4 ± 173.5	132.3 ± 158.9	165.9 ± 178.9	0.007 * (−0.195)
Caffeine (mg·kg^−1^·day^−1^)	2.48 ± 2.85	1.82 ± 2.20	2.78 ± 3.06	<0.001 * (−0.338)
Caffeine (mg·day^−1^) #	172.5 ± 174.5	155.2 ± 161.5	179.8 ± 179.4	0.068 (−0.141)
Caffeine (mg·kg^−1^·day^−1^) #	2.75 ± 2.88	2.14 ± 2.23	3.01 ± 3.08	<0.001 * (−0.306)
Caffeine (*n* (%))	807 (91.1)	243 (87.4)	564 (92.8)	0.009 *
Coffee (*n* (%))	474 (53.5)	131 (47.1)	343 (56.4)	0.010 *
Instant coffee (*n* (%))	118 (13.3)	26 (9.4)	92 (15.1)	0.019 *
Tea/mate (*n* (%))	300 (33.9)	55 (19.8)	245 (40.3)	<0.001 *
Cola drinks (*n* (%))	274 (30.9)	83 (29.9)	191 (31.4)	0.641
Energy drinks (*n* (%))	95 (10.7)	50 (18.0)	45 (7.4)	<0.001 *
Chocolate (*n* (%))	496 (56.0)	145 (52.2)	351 (57.7)	0.121

Values are the mean ± SD, or number of participants (percentage) consuming caffeine and caffeinated products. # Mean values among caffeine consumers (*n* = 807, men *n* = 243, women *n* = 564). * *p* < 0.05 indicates significant differences between men and women, as determined by Student’s *t*-test for unpaired data or Pearson’s chi-square (χ2). Cohen’s d is provided as a measure of effect size for *t*-test comparisons.

**Table 3 nutrients-14-01661-t003:** Sources of caffeine intake among participants in the study.

	All (*n* = 807)	Men (*n* = 243)	Women (*n* = 564)	*p* Value (Cohens’ d)
Coffee (%)	48.9 ± 42.7	47.1 ± 43.9	49.6 ± 42.2	0.460 (−0.058)
Instant coffee (%)	3.9 ± 14.0	3.2 ± 14.0	4.5 ± 14.0	0.625 (−0.038)
Tea/mate (%)	15.7 ± 28.9	9.25 ± 24.0	18.4 ± 30.4	<0.001 * (−0.320)
Cola drinks (%)	7.8 ± 21.9	9.3 ± 24.4	7.2 ± 20.8	0.248 (0.096)
Energy drinks (%)	3.4 ± 13.9	7.6 ± 21.6	1.6 ± 8.2	<0.001 * (0.444)
Chocolate (%)	20.3 ± 36.6	23.1 ± 38.8	19.1 ± 35.6	0.170 (0.110)

Values are expressed as means ± S.D. and represent the caffeine contribution in percentage of each source with respect to total caffeine intake. * *p* < 0.05 indicates significant differences between men and women, as determined by Student’s *t*-test for unpaired data. Cohen’s d is provided as a measure of effect size.

**Table 4 nutrients-14-01661-t004:** Multivariate regression analysis for daily caffeine intake (mg·day^−1^).

Variable	B	β	95%CI	t	*p* Value	R^2^	Adjusted R^2^	R^2^ Change
Smoking	85.756	0.172	52.799, 118.714	5.107	<0.001 *	0.052	0.051	0.052
Fruit/Vegetables	12.394	0.149	7.190, 17.598	4.675	<0.001 *	0.078	0.076	0.026
Alcohol	57.734	0.141	31.699, 83.769	4.352	<0.001 *	0.100	0.097	0.023
Cannabis	91.897	0.109	36.670, 147.124	3.266	0.001 *	0.111	0.107	0.010
Sex	29.271	0.078	5.844, 52.699	2.452	0.014 *	0.117	0.112	0.006
Age	5.524	0.066	0.285, 10.763	2.069	0.039 *	0.121	0.115	0.004

Model: *p* < 0.001 (ANOVA). B: regression coefficient; β: standardized beta coefficient. The positive coefficients for smoking, alcohol and cannabis indicate higher values for caffeine intake in consumers of these substances than in non-consumers. The positive coefficient for sex indicates higher values for caffeine intake in women than in men. * *p* < 0.05 indicates significant predictors or R^2^ changes.

**Table 5 nutrients-14-01661-t005:** Logistic regression analysis for caffeine intake (consumers and non-consumers).

Variable	OR Adjusted	95%CI	*p* Value
**Sex** (Reference men)	1.750	1.085, 2.822	0.022 *
**Age**	1.064	0.942, 1.202	0.320
**Smoking** (Reference non-smokers)	3.359	1.003, 11.526	0.048 *
**Cannabis consumption** (Reference non-consumers)	2.058	0.262, 16.171	0.493
**Alcohol consumption** (Reference non-consumers)	2.047	1.252, 3.346	0.004 *
**Daily fruit and vegetable servings**	1.121	0.989, 1.271	0.073

Reference: non-consumers of caffeine; * *p* < 0.05 indicates significant odds ratios (OR).

**Table 6 nutrients-14-01661-t006:** Caffeine and coffee consumption in participants distributed per smoking habit.

	Non-Smokers (*n* = 706)	Ex-Smokers (*n* = 55)	Occasional Smokers (*n* = 52)	Habitual Smokers (*n* = 73)	ANOVA
Caffeine (mg·day^−1^)	137.0 ± 165.4	169.4 ± 165.7	250. 9 ± 201.2	254.3 ± 181.1	*p* < 0.001 * (η^2^ = 0.054)
Coffee (servings·day^−1^)	0.95 ± 1.34	1.20 ± 1.35	1.92 ± 1.69	2.04 ± 1.72	*p* < 0.001 * (η^2^ = 0.063)

Values are expressed as means ± S.D. * *p* < 0.05 Indicates significant differences (ANOVA one-way test). η^2^ value is provided as a measure of effect size.

**Table 7 nutrients-14-01661-t007:** Caffeine intake and prevalence of consumption of caffeine and caffeinated products in smoker and non-smoker participants.

	Non-Smokers (*n* = 761)	Smokers (*n* = 125)	*p* Value (Cohen’s d)
Caffeine (mg·day^−1^)	139.3 ± 165.5	252.9 ± 189.0	<0.001 * (−0.672)
Coffee (servings·day^−1^)	0.99 ± 1.34	1.99 ± 1.70	<0.001 * (−0.729)
Caffeine (*n* (%))	685 (90.0)	122 (97.6)	0.006 *
Coffee (*n* (%))	378 (49.7)	96 (76.8)	<0.001 *
Instant coffee (*n* (%))	98 (12.9)	20 (16.0)	0.341
Tea / mate (*n* (%))	260 (34.2)	40 (32.0)	0.635
Cola drinks (*n* (%))	215 (28.3)	59 (47.2)	<0.001 *
Energy drinks (*n* (%))	70 (9.2)	25 (20.0)	<0.001 *
Chocolate (*n* (%))	431 (56.6)	65 (52.0)	0.333

Values are the mean ± SD, or number of participants (percentage) consuming caffeine and caffeinated products. *p* < 0.05 * indicates significant differences between smokers and non-smokers, as determined by Student’s *t*-test for unpaired data or Pearson’s chi-square (χ2). Cohen’s d is provided as a measure of effect size for *t*-test comparisons.

**Table 8 nutrients-14-01661-t008:** Multivariate regression analysis for subjective sleep quality.

Variable	B	β	95%CI	t	*p* Value	R^2^	Adjusted R^2^	R^2^ Change
Caffeine	0.004	0.194	0.003, 0.006	5.839	<0.001 *	0.034	0.033	0.034
Sex	0587	0.072	0.058, 1.117	2.178	0.030 *	0.050	0.046	0.005
Age	−0.079	−0.043	−0.198, 0.040	−1.299	0.194			
Fruit/Vegetables	−0.197	−0.108	−0.315, −0.078	−3.248	0.001 *	0.045	0.042	0.010
Smoking	0.321	0.030	−0.429, 1.072	0.840	0.401			
Cannabis	−0.821	−0.044	−2.079, 0.438	−1.280	0.201			

Regression model: *p* < 0.001 (ANOVA). B: regression coefficient; β: standardized beta coefficient. Caffeine: caffeine intake (mg/day); fruit/vegetables: servings of daily fruit and vegetables intake. The positive coefficient for sex indicates higher values of MOS Sleep in women than in men. * *p* < 0.05 indicates significant predictors or R^2^ changes.

**Table 9 nutrients-14-01661-t009:** Results of the Caffeine Motives Questionnaire.

CMQ (Score Range)	All (*n* = 807)	Men (*n* = 243)	Women (*n* = 564)	*p* Value (Cohen’s d)
CMQ Global (21–105)	37.21 ± 9.14	36.47 ± 9.08	37.50 ± 9.16	0.262 (0.112)
CMQ Factor 1-Cognitive enhancement (6–30)	15.31 ± 6.18	14.91 ± 5.94	15.47 ± 6.27	0.363 (−0.091)
CMQ Factor 2-Negative affect relief (3–15)	4.06 ± 2.10	3.96 ± 2.01	4.10 ± 2.14	0.508 (−0.066)
CMQ Factor 3-Reinforcing effects (9–45)	14.47 ± 3.91	14.39 ± 4.02	14.50 ± 3.87	0.784 (−0.027)
CMQ Factor 4-Weight control (3–15)	3.38 ± 1.05	3.22 ± 0.78	3.44 ± 1.14	0.016 * (−0.206)

CMQ: Caffeine Motives Questionnaire. * *p* < 0.05 Indicates significant differences between men and women, as determined by Student’s *t*-test for unpaired data. Cohen’s d is provided as a measure of effect size.

## Data Availability

The data presented in this study are available on request from the corresponding author.

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
