# Peer review of "Caffeine Intake among Undergraduate Students: Sex Differences, Sources, Motivations, and Associations with Smoking Status and Self-Reported Sleep Quality"

_nutrients, 2022, doi:10.3390/nu14081661_

Round 1
Reviewer 1 Report
The current study examined predictors of self-reported caffeine consumption and sleep quality in a sample of about 900 undergraduate students in Spain. Predictors of interest included sex, age, smoking, and alcohol, cannabis, and fruit and vegetable consumption; and caffeine additionally for self-reported sleep quality. Also investigated were sources of and motivations for caffeine consumption. Findings indicated that more smoking, consumption of fruits, vegetables, alcohol, and cannabis, female sex, and higher age were associated with more caffeine consumption. More caffeine, female sex, and less fruit/vegetable consumption were associated with poorer self-reported sleep quality. The greatest source of caffeine was coffee. The most relevant motivation for caffeine consumption was the desire for cognitive enhancement. The topic of caffeine and sleep quality is important, and the current study provides some new knowledge. I have some concerns and suggestions, mainly regarding the novelty and the examination of sex differences.
Major concerns and suggestions
- The authors have not made clear why the current study is novel or needed. Several studies exist that examine motivations for caffeine consumption, the association of caffeine with smoking, and the association of caffeine with self-reported sleep quality.
- The manuscript focuses much on sex differences, yet the title does not mention sex. There is no literature review on sex differences in caffeine, sleep quality, fruit/vegetable intake, or substance use; nor is there justification for examining sex differences.
- Given the examination of sex differences, it would be prudent to examine whether sex moderates the association between caffeine and smoking, and between caffeine and sleep quality (perhaps as supplementary, given the large amount of results already presented).
Moderate concerns and suggestions
- No hypotheses are provided in the Introduction.
- Please specify "self-reported" or "subjective" every time you mention sleep quality as measured in the current study.
- The Methods, Results, and Discussion mention examining the associations of caffeine intake with fruit and vegetable consumption, yet these are not mentioned in the Introduction (literature review or aims).
- More information should be provided about the sample. How many individuals had access to Moodle, within the entire university? How many individuals could view the study on Moodle? Were the individuals who viewed the study on Moodle representative of the university as a whole? If not, how did they differ (race, age, socioeconomic status, major/area of study, etc.)?
- It's mentioned that 32 participants were excluded due to incomplete questionnaires. Which items were they specifically missing? Were these items related to the central aims of the paper? If they were only missing data that were irrelevant to the study aims, they should be included.
- "A minimum of 623 were required for significance" refers to which variables? Each association ostensibly has a unique effect size, meaning the number required for significance should differ among examined associations.
- Are there meaningful variations in race, foreign-born status, and/or ethnicity in the Balearic Islands? If so, were these assessed and controlled for?
- Was a measure of socioeconomic status assessed and controlled for? If not, please include as a limitation.
- Please include the exact question and frequency options provided to participants for caffeine intake.
- Similarly, please specify the questions used for alcohol intake and fruit/vegetable consumption, including what counted as a "serving."
- Were other measures of food intake, such as breakfast consumption, fast food, or sweetened drinks, assessed?
- Please explain the sentence "While this Factor 1 reached about 39% of the score 271 range, the other factors reached between 2.5% and 15.2% of the score range" in the Results section. Could you include a table with the most relevant factors, and the least relevant, by sex?
- For t-tests in between-sex analyses, please include Cohen's d as a measure of effect size, preferably in the table. Please also include a measure of effect size for the ANOVA presented in Table 6, such as Eta squared, partial Eta squared, or omega squared.
- Could the authors please speculate on the following: Why were consumption of fruits and vegetables, alcohol, and cannabis associated with more caffeine consumption? Why was smoking, a stimulant, not associated with poorer sleep quality?
- Please include the factor titles (or an abbreviated version of each) in the first column of Table 9 rather than simply the factor numbers.
- What are the implications of the findings?
Other concerns and suggestions
- There are some grammatical errors in the manuscript. For example, "drug" should be "drugs" in the first sentence of the abstract; "reported" should be "report" at line 17; "the pattern" should be "patterns" in the third sentence of the abstract; "the associated use" should be "associations" in the same sentence; "take coffee/caffeine" should be "consume coffee/caffeine" throughout the paper; "Bonferroni corrections" should be "Bonferroni correction" in the fifth sentence of section 2.3; "associated to women" and "was associated with being women" should be "associated with female sex" in the paragraph at the end of page 6; "being men" should be "male sex" in the next sentence and at the end of page 7, "to the one found" should be "to that found" at the beginning of page 9, "than the recommended" should be "than the recommended amount" at the beginning of page 9; "or" should be "of" at line 350; "its" should be "their" at line 366; "These results could agree with the ones" should be "This is similar to the findings" at line 374; the sentences beginning with "in this regard" at line 399; "It is noteworthy that" at lines 322 and 409; and "the different tools used" at line 419 should be reworded, etc. Please carefully proofread the paper to ensure correct word use, subject-verb agreement, sentence structure, and clarity.
- The abstract lacks an implications sentence at the end.
- I suggest changing the title to "Caffeine intake among undergraduate students: Sex differences, sources, motivations, and associations with smoking status and self-reported sleep quality." The current title is difficult to follow, and it is important to specify that sleep quality is self reported.
- The Introduction describes literature on the association of objectively measured sleep quality with caffeine, which is not relevant to the current analyses that use self-reported sleep quality. Please instead include studies with self-reported quality.
- Relatedly, the description of the association of caffeine with restlessness and anxiety in the Introduction is not relevant to the current study.
- The sections in the Introduction describing the prevalence of caffeine consumption could be streamlined and shortened, and a more thorough literature review on the link between smoking and caffeine, and sleep quality and caffeine, should be provided.
- Please improve the flow of the Introduction and Discussion, which currently skip from topic to topic, even within the same paragraph. Each paragraph should have a topic sentence, and then the remaining sentences should expand upon that topic. New topics should be introduced in new paragraphs.
- "The aim of this study was to determine the pattern of caffeine consumption" is vague. Please specify what patterns were examined.
- I suggest use of the Oxford comma to improve clarity.
- Please keep consistent tense when describing results. For example, use past tense when describing specific studies and what was accomplished in the current study, and present tense when describing general findings.
- "University" and "undergraduate" are somewhat redundant. I suggest only using "undergraduate."
- In the Results, I suggest translating caffeine mg amounts into more understandable units – e.g., 172.5 mg caffeine is about two cups of coffee.
- Please reword the title "3.3. Multivariate linear and logistic regression analysis of caffeine and caffeine sources’ consumption" to make clearer the topics of the section.
- Please clarify in the first sentence of section 3.3 that the linear regression examined mg of caffeine, and the logistic regression examined whether an individual consumed caffeine or not.
- Please re-introduce the study aims at the beginning of the Discussion section.
- In the Discussion, is sometimes difficult to ascertain when the authors are discussing their own results, and when they are referring to previous studies. Please make sure this distinction is clear, especially by avoiding words such as "this" without identifiers.
- It may be worth mentioning a recent relevant study published in the same journal which also found an association between caffeine consumption and poorer self-reported sleep quality, both measured through daily diary (doi:10.3390/nu14010031).
Reviewer 2 Report
Dear Athors,
Congratulations on your research on the caffeine intake. It is very important to control and evaluate caffeine intake, especially in young people. The study was conducted correctly using the validated Caffeine Motives Questionnaire (CMQ on a sample of 886 undergraduate students).
Below are my suggestions / comments:
The sampling was not clearly described. The research was carried out within one or many universities?
The dependent variable (regressant) was not described in linear regression. What is the role of the variable "sex" in this model? The variable "sex" is nominal.
Similarly, in the case of logistic regression, please provide brief information on the dependent variable.
Why were two models (multivariate regression and logistic regression) used to predict caffeine intake?
More information is needed on sleep quality variable (regressant) in the next model.
Reviewer 3 Report
This paper describes the outcomes of a study examining patterns of caffeine consumption and its association with demographics and health behaviors in university students. The paper is clearly written and the study methodology well explained. My main concern with this paper is that it is not clear what these data are adding to the literature other than confirming findings from a variety of previous studies with university students that the authors have cited. What is the gap in our knowledge being addressed with these findings? The aim of the study is simply stated as “to determine the pattern of caffeine consumption among undergraduate students from the University of the Balearic Islands.” If the findings from previous studies with university students do not apply to the students in the present study, this should be clarified in the text. Otherwise, the authors need to expand on the goals of this work and the questions it sought to address.
Additional Minor Comments:
-The supplemental tables are not referenced in the text.
-Line 260, “higher sleep quality” is confusing because a few sentences earlier it says that “higher values of the scale being associated with a worse sleep quality”. Consider saying “better sleep quality” to avoid the confusion with the direction of the scale.
Round 2
Reviewer 1 Report
Thank you for providing this detailed revision in response to the reviewer comments. It is evident the authors put in much work and improved the manuscript substantially. I have some concerns I do not believe were sufficiently addressed in the revision, and/or some requests for clarification (below):
- The authors mention that smoking was a more important predictor of caffeine consumption in men than in women, but this does not seem to be borne out by the results presented in Tables S1 and S2. Smoking predicted caffeine intake in both sexes, p < .001. The R2 for smoking was much higher in men, but to declare these associations meaningfully different, one would have to examine the sex moderation (as I previously recommended) – whether sex significantly interacts with smoking on caffeine consumption (interaction term sex * smoking). This was the nature of my original suggestion. Did the authors do this? If so, please include the results of these analyses (suggest in Tables S1 and S2 if possible). If not, the authors should not declare these associations as different between the sexes. I suggest conducting these analyses.
- The authors' explanation for exclusion of hypotheses is not justified. Hypotheses that are defined before the analyses are performed (a priori) are necessary to reduce Type I error. There may be some areas that are more exploratory that do not require hypotheses (such as the association between fruit and vegetable consumption and caffeine intake), but it is clear that certain associations are well established in the literature (e.g., caffeine and poor sleep quality), and hypotheses should be formulated based on these previous studies. It is also explicitly stated in the Nutrients author guidelines: "It [the Introduction] should define the purpose of the work and its significance, including specific hypotheses being tested."
- The authors stated that they clarified that convenience sampling was used, but this is not mentioned in the manuscript. It is also unclear, as mentioned in the original review, whether the collected sample was representative of the university. The authors may have aimed to contact a representative sample, but that does not mean they achieved one. It is necessary for readers to have a sense of whether these results can extrapolate to the whole university, and, by extension, possibly to other Spanish undergraduate students.
- Please be more specific about how many participants were missing items related to smoking, cannabis, and alcohol consumption, for each of these variables. In other words, specify exactly how the sample size was arrived at, and why these 40 participants were excluded.
- The author's explanation of how they chose the "minimum 623 required for significance" requires clarification. What is "the number of participants required for descriptive purposes"? Do you mean a large enough number to be generalizable to the undergraduate students from the University of the Balearic Islands? If so, please state this plainly. Please also explain exactly how the 623 number was arrived at – what calculations were performed?
- A minor suggestion: Make font style consistent for the headers of Table S1 and S2 (Palatino Linotype, instead of Calibri) in the Supplementary Information.
Reviewer 3 Report
I greatly appreciate the authors' expansion on the research question and originality of these findings. I think the expanded justification in the Introduction has made the manuscript much stronger.
Two very minor additional comments:
-Lines 157-158 present the options participants were given in response to the question of frequency of alcohol consumption. Two of the options presented are 2-4 times a week and 2-3 times a week. If this is a typing error, it should be corrected. Otherwise, it is unclear how these options are different and how participants would decide which describes them.
-Line 428, “hypothesis” is spelled incorrectly.
